# Channel Emulator Framework for Underwater Acoustic Communications

**Indrakshi Dey** [1,2,*] and **Nicola Marchetti** [2]

1   Walton Institute of Information and Communication Sciences, X91 P20H Waterford, Ireland
2   School of Engineering, Trinity College Dublin, D02 PN40 Dublin, Ireland; nicola.marchetti@tcd.ie
*   Correspondence: indrakshi.dey@waltoninstitute.ie or deyi@tcd.ie

**Abstract:** In this paper, we develop a tractable mathematical model and an emulation framework for communicating information through water using acoustic signals. Water is considered one of the most complex media to model due to its vastness and variety of characteristics, which depend on the scenario, the type of water body (lakes, rivers, tanks, sea, etc.), and the geographical location of the water body being considered. Our proposed mathematical model involves the concept of damped harmonic oscillators to represent the medium (water); Milne's oscillator technique is used to map the interaction between the acoustic signal and water. Wave equations formulated for acoustic pressure and acoustic wave velocity are employed to characterise the travelling acoustic signal. The signal strength, phase shift, and time delay generated from the mathematical model are then inputted into a Simulink-based emulator framework to generate channel samples and channel impulse responses. The emulator utilises the wide sense stationary uncorrelated scattering (WSSUS) assumption and a finite sum-of-sinusoids (SOS) approach with a uniformly distributed phase to generate the channel samples. By utilising this emulator platform, it becomes feasible to generate profiles for amplitude variation, the Doppler shift, and spread experienced by any travelling signal in various underwater communication scenarios. Such a platform can be employed to simulate different communication scenarios, underwater network topologies, and data for training various learning models. Additionally, it can predict the performance of different modulation, multiplexing, error correction, and multi-access techniques for underwater acoustic communication (UWAC) systems.

**Keywords:** acoustic signals; underwater communications; channel impulse responses; Doppler frequencies; performance evaluation and monitoring; Simulink-based emulator



## 1. Introduction

Emerging applications, including water quality monitoring [1], offshore asset monitoring [2], biodiversity monitoring [3], oil field exploration [4], biogeochemical process exploration [5], and the extensive deployment of acoustic modems with their efficient computing and denoising capabilities [6], have attracted significant investments of time, finances, and intellectual resources from both industry and academia in recent years. These investments aim to develop viable underwater acoustic communication (UWAC) systems. However, the complex nature of water, the slow propagation speed of acoustic waves through water, limited bandwidth availability, complex noise sources, and delay-Doppler effects make it much more challenging to design and deploy a reliable UWAC system compared to traditional terrestrial radio systems [7,8]. The only way forward is to design dedicated signal processing algorithms and network protocols specifically for UWAC scenarios, which are validated across a variety of scenarios.

Running actual experiments in seas, rivers, or lakes presents significant financial and logistical challenges [9]. Border security protocols among multiple nations sharing a water body can sometimes make it nearly impossible to conduct trials, especially during the design stage. Therefore, it is crucial for system engineers to test their designs using emulator platforms that are both controllable and offer realistic and comprehensive simulations.

Channel emulator platforms allow designers to evaluate the performance of different algorithms across a broad range of parameters. They also provide ample samples to assess how the system performs and evolves over time through thorough empirical evaluations [10]. Once the robustness of the designed system has been synthetically validated, experimental demonstrations can be arranged in a specific environment with significantly reduced time, financial, and logistical commitments.

Developing channel emulators for the UWAC environment poses significant challenges due to the absence of a typical underwater environment. Water characteristics vary based on factors such as depth, salinity, density, and geographical location. The acoustic signal is subjected to fluctuations in attenuation, absorption, delay, Doppler effects, and noise, influenced by the diverse underwater characteristics, as well as sea-surface and sea-bottom contours [11,12]. Therefore, instead of separately characterising each environmental parameter and its impact, we propose the development of a mathematical model that utilises the principles of the medium's physics and acoustic signal propagation. This model aims to accurately map the interaction between the aquatic medium and the travelling signal.

Several UWAC channel modelling software and platforms have been developed over the years to realistically represent different UWAC communication scenarios. One very popular open-source platform is BELLHOP [13], which utilises beam tracing to emulate acoustic pressure fields in specific underwater environments. Another widely used platform is the Ulrick model [14], which calculates distance-related spreading loss and frequency-related absorption loss for UWAC links. Other UWA network simulators, such as DESERT [15], SUNSET [16], and WOSS [17], also provide researchers with valuable insights into the UWAC environments. However, there is no single unified platform that comprehensively captures the flow of the acoustic signals through various aquatic environments. Such a platform would provide valuable insights into how the signal interacts with the environment, how it evolves over time and distance, and how these changes impact the overall design of the communication system.

The primary contribution of this paper is the integration of mathematical modelling, specifically how acoustic signals propagate through water, with the generation of channel samples using the sum-of-sinusoids (SOS) approach [18] in a generalised yet manageable channel emulator platform. The emulator is capable of characterising key aspects of the UWAC channel, such as the signal's amplitude profile, attenuation, propagation loss, delay, Doppler spread, and shift. These aspects are critical for designing signal processing techniques, networking protocols, and extracting network topology in UWAC systems. Specifically, we,

- Combine concepts of damped harmonic classical oscillators [19], Milne's oscillator [20], and acoustic wave equations to formulate a mathematical model for the propagation of acoustic signal through an underwater environment.
- Integrate outputs of the mathematical model with wide sense stationary uncorrelated scattering (WSSUS) and SOS-based models for the synthetic generation of channel impulse responses, signal amplitudes, phase change profiles, Doppler frequencies, and the Doppler power spectral profile.
- Develop a Simulink-based channel emulator platform that incorporates the oscillator-based mathematical model and WSSUS-based channel sample generation.
- Use the developed channel emulator platform, and demonstrate (i) snapshots of channel samples for different scenarios and parameters and (ii) the performance of two example communication systems—(a) the differential $M$-ary phase shift keying (D$M$PSK)-orthogonal frequency division multiplexing (OFDM) system with 1/2-rate Turbo coding for single-transmit–single-receive scenarios; (b) a D$M$PSK-OFDM system with a 1/2-rate Turbo coding for multiple-transmit–multiple-receive scenarios.

For some time now, it has been common practice to conduct measurement campaigns in specific scenarios for which underwater acoustic communications (UWAC) systems are designed. This trend can be attributed to several reasons; (i) UWAC is an emerging field in communications system design, and it is not as well-established as radio frequency

(RF)-based communication systems for terrestrial scenarios. (ii) There is no typical UWAC scenario due to the diverse characteristics of the underwater medium and its vastness. (iii) Acoustic communication is not a widely prevalent signal carrier form compared to other technologies. However, throughout the years, emulation platforms for underwater scenarios, such as BELLHOP, have emerged, which can reliably approximate the communication scenario. On the other hand, the idea behind our work is to understand the physics of acoustic waves travelling through water and represent the interaction in a way that is similar to Maxwell's equations. Maxwell's equations or empirical channel models (such as the Rayleigh fading model) are good enough for designing robust radio frequency (RF) communication systems. Ray-tracing techniques in RF communications also provide detailed maps of specific scenarios; however, they are rarely used when designing general systems. Similarly, we believe that our physics-based simulation platform would be good enough to design reliable UWAC systems.

## 2. Mathematical Modelling of Propagating Signal

### 2.1. Modelling the Medium

We begin with the damped classical harmonic oscillator equation [19],

$$\ddot{y} + \beta \dot{y} + \omega^2 y = 0 \tag{1}$$

where $y$ is the position, $\omega$ is the operating frequency, and $\beta$ is the linear damping coefficient. If varying the oscillator parameters offers modification to the classical oscillator, the harmonic oscillator equation in (1) is modified to the parametric oscillator representation. As the travelling acoustic signal interacts with water, the water movement (medium) can be modelled using the parametric oscillator equation,

$$\ddot{y} + \beta(t) \dot{y} + \omega^2(t) y = 0 \tag{2}$$

where the parameters $\beta$ and $\omega$ become functions of time, $\omega(t) > 0$ and $\omega(t) \to 1$ with $t \to \pm\infty$. Using constant-time scaling, the upper limit for $\omega(t)$ is set to 1. When there is no travelling signal, $\omega = 1$, i.e., the medium is an undriven oscillator state with unit frequency.

### 2.2. Modelling the Acoustic Signal

For signal modelling, we begin with the acoustic wave equation,

$$c^2 \frac{\partial^2 \mathbf{p}}{\partial x^2} = \frac{\partial^2 \mathbf{p}}{\partial t^2} \tag{3}$$

where $\mathbf{p}$ is the acoustic pressure, $x$ is the direction of the signal propagation, and $c = 1480$ m/s is the velocity of the acoustic wave through water. In (3), $\mathbf{p}$ can be expressed as

$$\mathbf{p} = f_1(y + ct) + f_2(y - ct) \tag{4}$$

with amplitude $\alpha$, wave number $k$ (where $k = 2\pi/\lambda$ and $\lambda$ is the signal wavelength, and rearranging (4),

$$y = \frac{1}{k} \cos^{-1}\left(\frac{\mathbf{p}}{\alpha}\right) + ct \tag{5}$$

and inserting (5) in (2), we obtain

$$\ddot{\mathbf{p}} \frac{\mathbf{p}^2}{\alpha^2} - \mathbf{p}(\dot{\mathbf{p}})^2 + \beta(t) \dot{\mathbf{p}} \frac{\mathbf{p}^2}{\alpha^2} - \beta(t) ck\mathbf{p} - \omega^2(t)\mathbf{p} \cos^{-1}\left(\frac{\mathbf{p}}{\alpha}\right) - \omega^2(t) ctk\mathbf{p} = 0. \tag{6}$$

If $y = ct$ satisfies (6), we can write the so-called Milne's equation [20] with $\mathbf{p} = \alpha$ to obtain

$$\ddot{\mathbf{p}} - \mathbf{p}(\dot{\mathbf{p}})^2 + \beta(t) \dot{\mathbf{p}} - \beta(t) ck\mathbf{p} - \omega^2(t) ctk\mathbf{p} = 0. \tag{7}$$

Here, (7) is the well-known Milne oscillator equation. Essentially, (7) represents the evolution of the acoustic pressure field with time and can be solved for **p** to calculate the effective receive signal strength (Milne energy), effective phase shift, and time-delay experienced by the travelling signal.

### 2.3. Modelling the Interaction between the Medium and the Acoustic Signal

The next step is to solve **p** using the Lagrangian of (7) for continuous media and fields with finite degrees of freedom:

$$\mathcal{L} = \frac{1}{2}\dot{\mathbf{p}}^2 + \frac{\beta(t)ck}{2}\mathbf{p}^2 + \frac{\omega(t)^2ctk}{2}\mathbf{p}^2 \tag{8}$$

where $\frac{1}{2}\dot{\mathbf{p}}^2$ is the kinetic energy density and $(\frac{\beta(t)ck}{2}\mathbf{p}^2 + \frac{\omega(t)^2ctk}{2}\mathbf{p}^2)$ is the potential energy density. The next step is to formulate the Hamiltonian of (8) using the Euler–Lagrange concept to obtain

$$\mathcal{H}(\dot{\mathbf{p}}, p, t) = \dot{\mathbf{p}}\frac{\partial\mathcal{L}}{\partial\dot{\mathbf{p}}} - \mathcal{L} = \frac{1}{2}\dot{\mathbf{p}}^2 - \frac{\beta(t)ck}{2}\mathbf{p}^2 - \frac{\omega(t)^2ctk}{2}\mathbf{p}^2. \tag{9}$$

The difference between kinetic and potential energies, as reflected in Equation (9) remains constant before and after the duration over which the signal exists. Thus, the Hamiltonian, denoted as $\mathcal{H}$, can be expressed as the asymptotic Milne energy, $E_M$, rather than the effective signal strength at any point between the transmit and receive points on a UWAC link.

Here, we introduce the concept of path bundles, **q**, which consists of the solution to the Hamiltonian in (9) bounded by $E_M \geq 1$. Physically, the vector **q** represents the transmission rays that travel along a specific direction. The corresponding Milne energy can be expressed as,

$$E_M = \frac{1}{2}\dot{\mathbf{q}}^2 - \frac{\beta(t)ck}{2}\mathbf{q}^2 - \frac{\omega(t)^2ctk}{2}\mathbf{q}^2 \tag{10}$$

By solving for **q** in (10), we can express **q** as follows:

$$\mathbf{q} = \sqrt{\mathbf{q}_\pm^2 \cos^2(t-\tau) + \mathbf{q}_\mp^2 \sin^2(t-\tau)} \tag{11}$$

where **q** physically represents the amplitude of the acoustic ray bundles travelling through water (the path bundles include both the direct rays and reflected rays that constitute the total acoustic signal), and $\tau$ is the effective time delay experienced by the travelling signal. The derivative of **q** in (11) is set to $\dot{\mathbf{q}} = 0$. Putting $\dot{\mathbf{q}} = 0$ back in (10),

$$E_M = -\frac{\beta(t)ck}{2}\mathbf{q}^2 - \frac{\omega(t)^2ctk}{2}\mathbf{q}^2. \tag{12}$$

The above equation (12) can be solved to obtain

$$\mathbf{q} = \pm\imath\sqrt{2E_M/(\beta(t)ck + \omega(t)^2ctk)}$$
$$\mathbf{q}(t) = \pm\imath\sqrt{2E_M\cos(2t-\tau)/(\beta(t)ck + \omega(t)^2ctk)} \tag{13}$$

where $\imath = \sqrt{-1}$. Figure 1 demonstrates the evolution of the amplitude of the acoustic path bundles (**q**) over time for different values of the damping constant $\beta$. It is important to note that we did not vary $\beta$ as a function of time $t$. For our simulations, we kept $\beta$ fixed over the time period in which the amplitude evolution of the signal path bundles was observed. The amplitude evolution of the signal paths was formulated using (12).

Finally, the interaction between **q** and water over time can be represented in the form of the transition matrix, $\mathbf{M_q}$, where

$$\mathbf{M_q} = \mathbf{D}\begin{pmatrix} \cos 2\delta & \mathbf{q}_-^2 \sin 2\delta \\ -\mathbf{q}_+^2 \sin 2\delta & \cos 2\delta \end{pmatrix}\mathbf{D}; \mathbf{D} = \begin{pmatrix} \cos\tau & -\sin\tau \\ \sin\tau & \cos\tau \end{pmatrix} \tag{14}$$

with $\delta$ being the effective phase shift experienced by the acoustic signal. With the value of **D**, $\mathbf{M_q}$ can be expressed as,

$$\mathbf{M_q} = \begin{pmatrix} \cos\tau\cos\delta + \mathbf{q}_-^2\sin\tau\sin\delta & \mathbf{q}_-^2\sin\delta\cos\tau - \sin\tau\cos\delta \\ \sin\tau\cos\delta - \mathbf{q}_+^2\sin\delta\cos\tau & \cos\delta\cos\tau + \mathbf{q}_+^2\sin\tau\sin\delta \end{pmatrix} \tag{15}$$

where

$$\mathbf{q}_-^2 = \frac{2E_M\cos(2t-\tau)}{\beta(t)ck + \omega(t)^2ctk} \tag{16}$$

$$\mathbf{q}_+^2 = -\frac{2E_M\cos(2t-\tau)}{\beta(t)ck + \omega(t)^2ctk} \tag{17}$$

Using (14), (15), and (17), it is possible to generate a range of values for **q**, $\delta$, and $\tau$ to represent different underwater scenarios. With the underlying mathematical model representing the flow of the acoustic signal through any underwater environment formulated, the next step is to build the channel model.

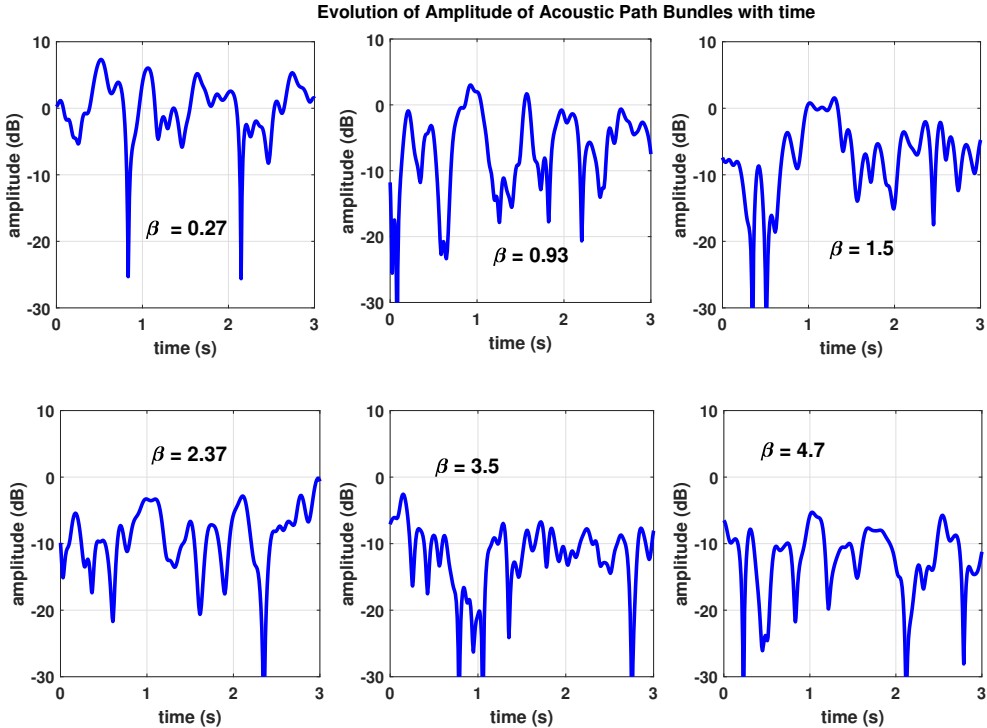

**Figure 1.** Evolution of the acoustic signal amplitude (energy) over time while travelling through water for different oscillator damping factors. The evolution of the acoustic amplitude is visualised through the representation using the concept of damped harmonic oscillators, Milne energy, and Milne oscillator.

### 2.4. Numerical Results

In Figure 2, we present a visualisation of the acoustic path bundles and reflections when they are travelling through water. We simulate three different scenarios where the transmitter and the receiver are located at different depths, heights, and distances from

each other. We selected three different scenarios, which are representative of scenarios with the transmitter at a height lower than the receiver (Scenario 1), with the transmitter and receiver at the same height (Scenario 2), and with the transmitter height higher than the receiver (Scenario 3). Scenario 1 involves the receiver being positioned 50 m above the waterbed and being higher than the transmitter, which is located 10 m from the bottom. They are situated 1 km apart from each other. In Scenario 2, both the transmitter and the receiver are positioned at the same height of 50 m from the bottom, and the total water depth is 500 m. The distance between the two communicating nodes is now 2 km. Scenario 3 involves the separation of the transmitter and the receiver by 5 km, with the transmitter positioned at a higher height than the receiver. However, if the transmitter is at a lower height than the receiver, numerous signal paths and reflections will be experienced. With an increase in depth, the echoes tend to disappear, rendering only a few weak direct signal paths. With an increase in the distance between the transmitter and the receiver, weak signal paths and a few reflections appear.

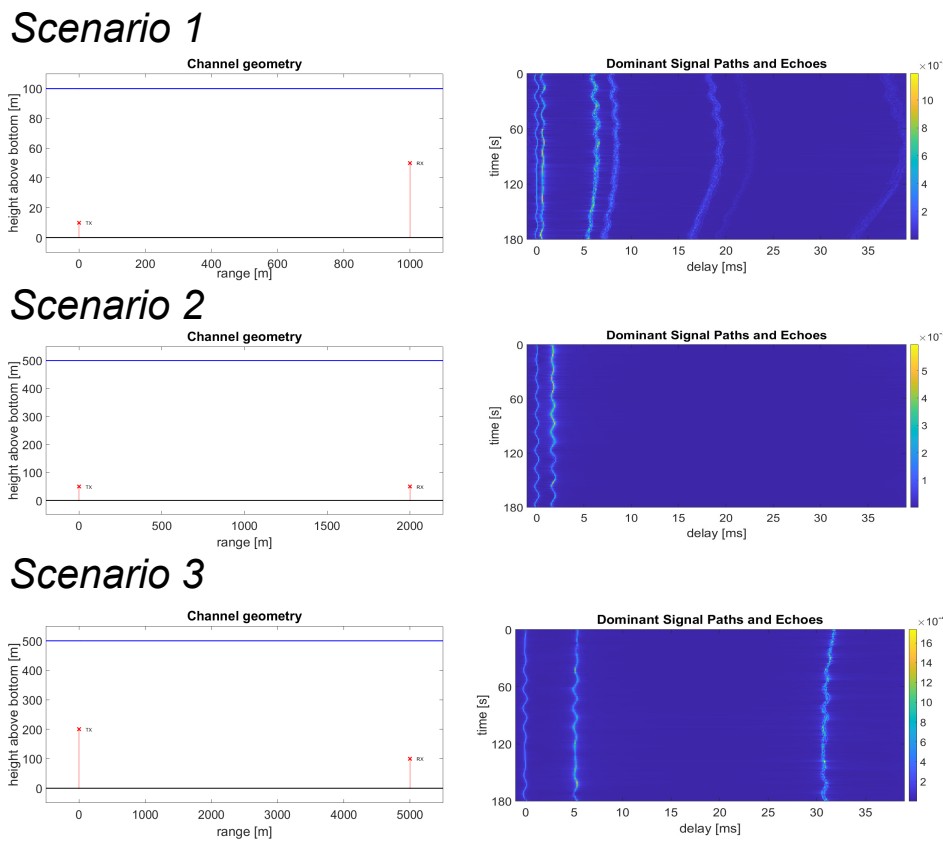

**Figure 2.** Visual representation of the acoustic signal path between a transmitter and a receiver located at different distances from the water body surface and different distances from each other. Scenario 1: Tx (transmitter) height = 10 m from the bottom and Rx (receiver) height = 50 m from the bottom, depth of water body = 100 m, the distance between Tx and Rx = 1 km. Scenario 2: Tx height = 50 m from the bottom and Rx height = 50 m from the bottom, depth of water body = 500 m, the distance between Tx and Rx = 2 km. Scenario 3: Tx height = 200 m from the bottom and Rx height = 100 m from the bottom, depth of water body = 500 m, the distance between Tx and Rx = 5 km.

## 3. Designing the UWAC Channel Emulator

Linear time-varying (LTV) systems are primarily used to characterise UWAC channels owing to their time-varying nature and varying environmental factors affecting the flow of acoustic signals through the water. When an acoustic signal travels through water, the signal is reflected by the surface and bottom of the sea, resulting in multiple delayed versions of transmission rays or path bundles. If we have $Q$ distinct reflected and refracted

rays (echoes) with amplitudes represented by $|\mathbf{M_q}|$ (where $|\cdot|$ represents the determinant of a matrix), and delays are denoted as $\tau_{\mathbf{q}}$, then the propagation channel with multiple information paths can be mathematically represented as follows:

$$\mathbf{M}(\tau, t, f) = \sum_{q=1}^{Q} |\mathbf{M_q}| \gamma_{\mathbf{q}}(f, t) \delta(\tau - \tau_{\mathbf{q}}(t)) \tag{18}$$

where $\gamma_{\mathbf{q}}(f, t)$ is the fading parameter characterising small-scale variations with each path bundle. If $\gamma_{\mathbf{q}}(f, t)$ is stationary in time, then we can use the time correlation function to derive the statistical description of $\gamma_{\mathbf{q}}(f, t)$,

$$\mathcal{R}_{\mathbf{q}}(f, \Delta t) = \mathrm{E}[\gamma_{\mathbf{q}}(f, t) \gamma_{\mathbf{q}}^*(f, t)] \tag{19}$$

where $\mathrm{E}[\cdot]$ denotes the expectation operation and $*$ denotes the complex conjugate. In this case, the Fourier transform of $\mathcal{R}_{\mathbf{q}}(f, \Delta t)$ yields the channel Doppler spectrum,

$$\mathcal{F}\{\mathcal{R}_{\mathbf{q}}(f, \Delta t)\} = S_{\mathbf{q}}(f, \eta) \propto e^{-\frac{|\eta|^\xi}{\chi}} \tag{20}$$

which can be approximated by the stretched exponential distribution. In (20), $\mathcal{F}\{\cdot\}$ denotes the discrete Fourier transform (DFT), $\eta$ represents the Doppler frequency, $\chi$ represents how closely the Doppler power spectrum matches the stretched exponential function, and $\xi$ denotes the stretching exponent. The time-domain autoregressive memory process-based generation of the stretched exponential spectrum is neither practical nor accurate in representation. This is because $\mathcal{R}_{\mathbf{q}}(f, \Delta t)$ has no closed form, and its Fourier transform is no longer exponentially distributed. Instead, in our developed platform, we generate random processes directly from the Doppler spectrum using the sum-of-sinusoids (SoS) model.

The delay power spectrum of the channel and its frequency correlation function can be expressed as

$$\mathcal{R}_{\mathbf{M}}(\tau) = \mathrm{E}[\mathbf{M}(t, \tau) \mathbf{M}^*(t, \tau')] = \sum_{q=1}^{Q} \mathrm{E}[|\mathbf{M_q}(t)|^2] \delta(\tau - \tau_{\mathbf{q}}(t)) \tag{21}$$

and

$$\mathrm{E}[\mathcal{F}\{\mathbf{M}(t, \tau')\} \mathcal{F}\{\mathbf{M}^*(t, \tau)\}] = \frac{1}{2\pi} \sum_{q=1}^{Q} \mathrm{E}[|\mathbf{M_q}(t)|^2] e^{j2\pi(f-f')\tau_{\mathbf{q}}} \tag{22}$$

respectively, assuming stationarity and uncorrelatedness. The difference between the initial frequency $f$ and the final frequency $f'$ defines the amount of the frequency correlation. Owing to the uncorrelated nature of $\tau_{\mathbf{q}}$, it is possible to calculate each path delay separately. We can then modify (22) in the delay domain to obtain

$$S(\tau, \eta) = \mathcal{F}\left\{ \sum_{q=1}^{Q} \mathrm{E}[\mathbf{M_q}(t) \mathbf{M}_{\mathbf{q}}^*(t + \Delta t)] \delta(\tau - \tau_{\mathbf{q}}(t)) \right\} \tag{23}$$

and from (23), we can obtain $S(\tau_{\mathbf{q}}, \eta)$, the delay-domain scattering of the $q$th path with respect to the Doppler spectrum.

Reflections due to the surface, bottom, surface–bottom, and surface–bottom–surface combination give rise to different sets of reflected paths with a range of Rician $K$-factors. Thus, we will use the SoS assumption to generate channel samples with amplitudes that are Rician-distributed. We consider the wide sense stationary uncorrelated scattering (WSSUS)

assumption here; therefore, we can represent channel samples $\mathbf{M}_q[i] = \mathbf{M}_q[iT_s]$ to formulate the SoS model:

$$\mathbf{M}[i] = \frac{1}{\sqrt{L(1+K)}}\left[\sum_{l=1}^{L} e^{j(\phi_l + 2\pi f_{dl} i T_S)} + \sqrt{K}e^{j(\phi_0 + 2\pi f_0 i T_S)}\right] \tag{24}$$

The time $t$ is sampled into intervals of $T_S$, where $L$ represents the number of path bundles for each tap. Within each tap, there are uniformly distributed random phases $\phi_l$ over the interval $[0, 2\pi]$, an initial phase $\phi_0$, and $L$ Doppler frequencies $f_d$. In (24), $K$ is the Rician $K$-factor, depending on the ratio of the signal power over the direct path to that over scattered paths and the direct line-of-sight (LoS) path acquires a Doppler frequency of $f_0$. It is evident that $L$ sets of Doppler frequencies $f_d$ can be selected from Jake's spectrum, such that these $f_d$s, in turn, can be used to generate (24), and the probability density function (pdf) of the range of $f_d$ can be expressed as,

$$S(\eta) = \frac{1}{2\chi}e^{-\frac{|\eta|^\xi}{\chi}}. \tag{25}$$

The values of $f_d$ can be formulated by applying the inverse transform sampling lemma [21] on (25). The inverse transform sampling lemma states that if the distribution function of input $X$ is given by $f(x)$, the cumulative distribution function (cdf), $F(x)$ will be uniformly distributed over the range $[0, 1]$. Consequently, input samples can be generated using $x = F^{-1}(\nu)$ where $\nu$ is uniformly distributed over the range $[0, 1]$, where $F$ denotes the distribution function. Using the above concept, the cdf of the Doppler frequencies $f_d$ can be expressed as,

$$\begin{aligned}
f_d &= F^{-1}\left[\int_{-\infty}^{f_d} \frac{1}{2\chi}e^{-\frac{|\eta|^\xi}{\chi}}\, \mathrm{d}\eta\right] = F^{-1}\left[\frac{1}{2\chi}\Gamma\left[(1/xi) - (-f_d/\chi)^\xi\right]\right] \\
&= -\chi\left[\sum_{\mu=0}^{1/\xi-1} (\mu+1)! W\left\{\frac{1}{\mu}\left(\frac{\mathrm{mod}(2u,1)\xi^2}{\xi-1}\right)^{1/\mu}\right\}\right]^{1/\xi}
\end{aligned} \tag{26}$$

where the obtained $f_d$s follows a stretched exponential distribution, mod refers to the modulo operation, and $W$ represents the product-Log or Lambert-$W$ function. It is worth-mentioning that the 'modulo' operation returns the remainder after a division of $2u$ by 1, and we consider that $u = F(f_d)$. A range of histograms of the $f_d$s using (25) and (26) is plotted in Figure 3. Using the discrete frequency-selective channel model, the channel impulse response is given by

$$|\mathbf{M}|(t;\tau) = \sum_{q=1}^{Q} E_{Mq}(t)\delta(\tau - \tau_q(t)) \tag{27}$$

where the term $E_{Mq}(t)$ represent the fading coefficients corresponding to the originally formulated Milne energy associated with each path bundle, and $\delta$ represents the delay function quantifying the delay experienced by the $q$th path. The delay is represented using the $\delta$ function as we are trying to model the infinite bandwidth system. In terms of system implementation, an infinite bandwidth system needs to transmit and receive filters for effective functioning.

Here, we consider a combination of a pulse-shaping transmit filter with an impulse response $g(t)$, the propagation channel $|\mathbf{M}|(t)$, and a matched receive filter, to represent a communication system with complex transmit samples $x_i$ and an output signal $z(t)$. If $z(t)$ is sampled at intervals that are multiples of $T$, or the sampling time $T_s$, the overall communication system will embody a discrete linear time-varying filter consisting of $n = 1, 2, \ldots, N$ filter taps. Therefore, such a communication system can be exemplified as a tapped delay line filter model. The filter taps, $h_n(\mu T_s)$, can be generated using digital

filtering that has a sampling rate capable of modelling a usable channel bandwidth for the channel model of (27).

These filter taps can be characterised by the following equation,

$$h_n[\mu] = \sum_{q=1}^{Q} \sqrt{\frac{|\mathbf{M}_q|}{L(1+K)}} \left[ \sum_{l=1}^{L} g(nT_s - \tau_q)e^{j(\phi_{l,q}+2\pi h_{l,q}(\mu T_s - \tau_q))} \right.$$
$$\left. + \sqrt{K}g(nT_s - \tau_0)e^{j(\phi_{l,q}+2\pi h_{l,q}(\mu T_s - \tau_0))} \right] \quad (28)$$

where $\tau_0$ is the initial delay and $|\mathbf{M}_q|$ are the average power levels observed over $Q$ path bundles. If we want to account for the drift velocity $v$ between the transmit and the receive terminals in an underwater environment, (28) is modified to

$$h_n[\mu] = \sum_{q=1}^{Q} \sqrt{\frac{|\mathbf{M}_q|}{L(1+K)}} \left[ \sum_{l=1}^{L} g((\mu - n + a\mu)T_s - \tau_q)e^{j(\phi_{l,q}+2\pi h_{l,q}(a\mu T_s - \tau_q))} \right.$$
$$\left. + \sqrt{K}g((\mu - n + a\mu)T_s - \tau_0)e^{j(\phi_{l,q}+2\pi h_{l,q}(a\mu T_s - \tau_0))} \right] \quad (29)$$

where $t + vt/c = (1+a)t$, $c \sim 1500$ m/s, and $a$ is the Mach number.

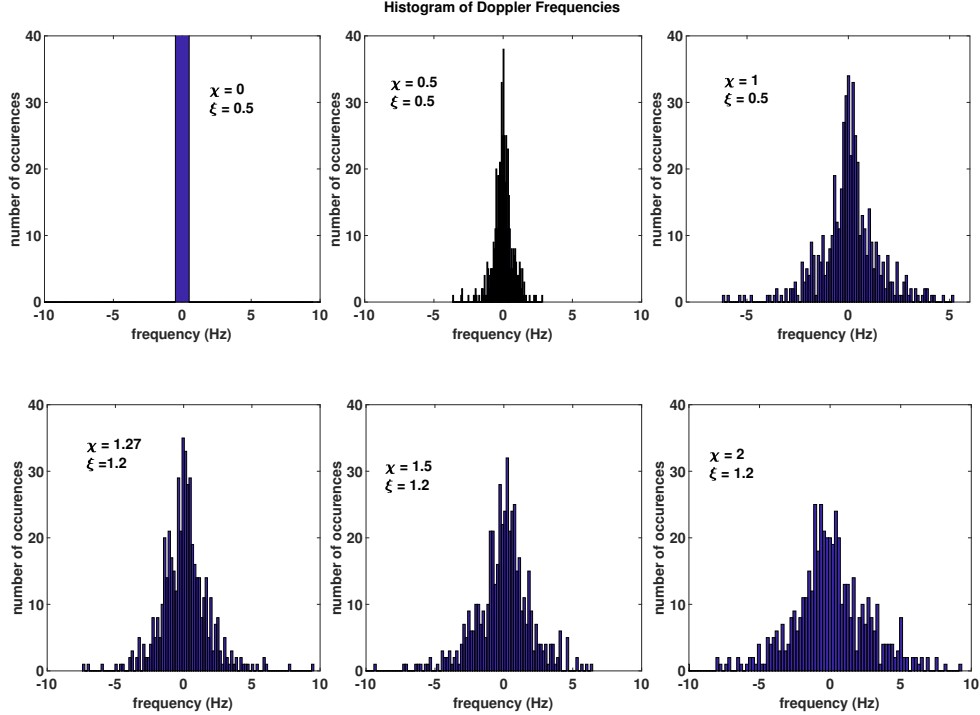

**Figure 3.** Histogram of Doppler frequencies generated using (25) and (26) for different stretching exponents, $\xi$, and different values of the closeness of the fit parameter, $\chi$.

The concept of the Mach number is introduced to represent the time-stretching or Doppler shift, i.e., the time period $t$ expands to $(t + vt/c)$. Here, $N$ is an important factor in the emulator design as it offers a trade-off between computational complexity and model stationarity. A sample set of trajectories generated over a 3-s interval and $L = 1000$, $K = 3$, $\xi = 1.2$ is presented in Figure 4. We also include the corresponding Doppler frequency distribution, Doppler power spectrum, and temporal coherence in Figure 4. Seldom do practical measurements from trials align with the WSSUS assumption for underwater propagation channels. However, in our work, we derive the amplitude of paths/echoes

of travelling sound waves, the delays experienced, and other relevant parameters directly from fundamental principles of physics. These concepts are derived from a mathematical description of how acoustic waves propagate through water and how water interacts with and influences the flow of acoustic waves. Therefore, it is possible to identify specific time periods and bandwidths within which the WSSUS assumption remains valid [22]. Accordingly, it is possible to determine an observation time $T_d$ in which the autocorrelation function of $|\mathbf{M}|(t, \tau)$, $\mathcal{R}_{\mathbf{M}}(t, \Delta\tau)$ is stationary with the constant mean value. Moreover, we can define an observation bandwidth $B_d$, where the frequency autocorrelation function, $\mathcal{R}_{\mathbf{M}}(f, \Delta t)$, is stationary.

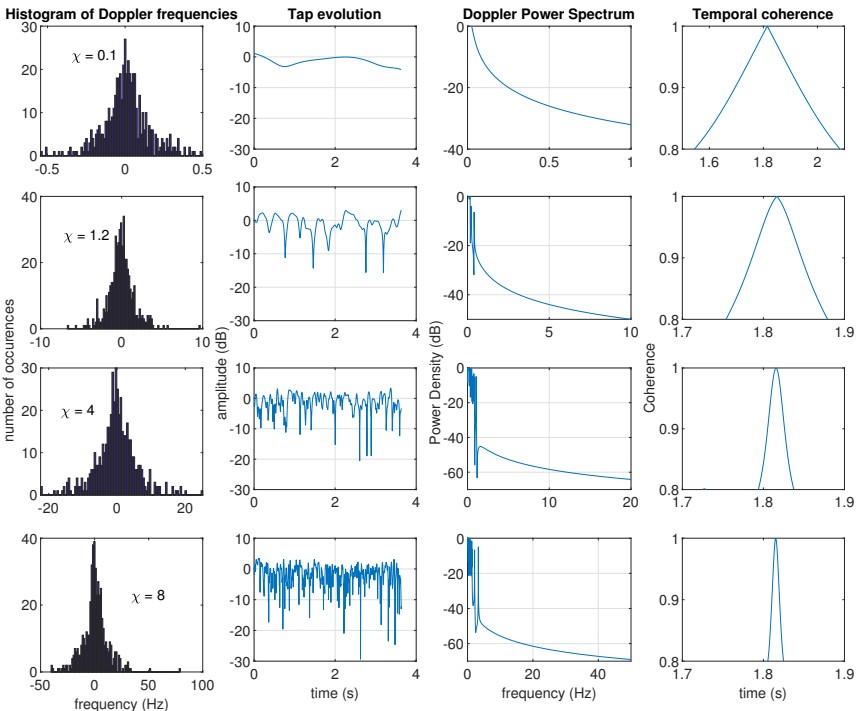

**Figure 4.** Sample histograms of Doppler frequencies, evolution of path amplitudes over 3 s, Doppler power spectrum, and corresponding temporal coherence generated using (18), (21), (26), and (29).

*Special Note on the Measurement Campaign*

We are currently in the process of collecting UWAC channel measurements in coastal and estuarine environments around Ireland and Wales as part of the STREAM project www.marinestream.eu. In STREAM, we are deploying underwater sensors to gather valuable information on temperature, rainfall, wind, marine nutrients, oxygen content, and phytoplankton abundance. However, as part of deploying underwater sensor networks, we are also collecting propagation channel information in the underwater environment for communication among acoustic sensors using acoustic signals. At the time of writing this paper, the measurement campaign is still ongoing and, thus, the results could not be reported in this paper. In the future, the model and emulator developed in this study will be validated using the measurements collected in the STREAM project.

## 4. Results and Discussion

Using (29), the channel representation was implemented in Matlab Simulink to develop the channel emulator. Separate Matlab functions were used to generate programmable path amplitudes $|\mathbf{M}_{\mathbf{q}}|$, delays $\tau_{\mathbf{q}}$, phase shifts $\phi_{l,q}$, the Doppler shift, and Doppler frequencies. A block diagram of the Simulink platform for the channel emulator is shown in Figure 5. The Simulink platform starts by incorporating the formulated Milne oscillator in (14) and (15) to generate the path amplitudes, path delays, and phase shifts. The platform also generates

Doppler shifts and frequencies using Matlab functions (M-functions) based on (25) and (26). Outputs of the Milne oscillator block and Doppler blocks are fed to the block of M-functions used for generating channel samples based on (27). The channel samples are then fed to the channel gain generator block to formulate the channel taps based on (28). The taps are then multiplied with the path delays obtained from the Milne oscillator-based model. We kept an additional (optional) noise model, which can be added for more realistic modelling. Multiple blocks are presented for each component to represent the generation of multiple channel taps.

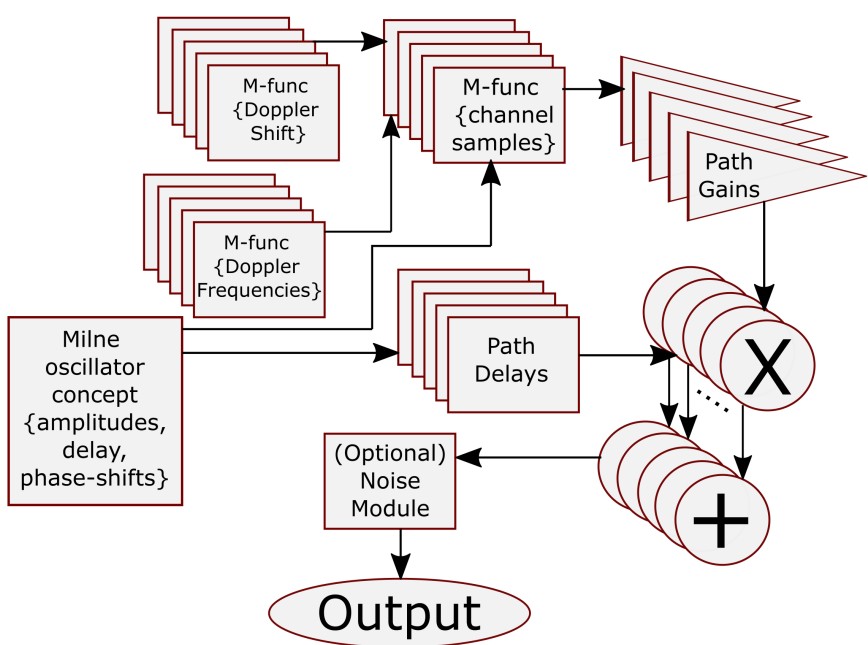

**Figure 5.** Block diagram of the Simulink model for the channel emulator. We have not yet included noise in our generated results; however, the emulator platform is very flexible and any noise model and distribution can be added (a generalization that we will explore in future work).

### 4.1. Simulation Results

The Simulink platform code can be easily adapted to the register transfer level (RTL) code and joint test action group (JTAG) co-simulations and can be ported to the field-programmable gate array (FPGA) platform for hardware implementation. The emulator possesses additional capabilities, such as data symbols, filtering, sampling, up-converting, and down-converting signals to be processed through the emulator. Example plots for channel responses over narrowband and wideband channels are provided in Figure 6.

In order to generate a simulated version of channel samples for different UWAC scenarios using our Simulink-based platform, we chose two different centre frequencies, i.e., 1 MHz (low) and 100 MHz (high), and two different channel bandwidths, i.e., 10 KHz (narrowband) and 20 MHz (wideband). Of these, a 1 MHz centre frequency with 10 KHz of bandwidth has already been established as the selectable range for UWAC [23] (refer to Figure 6 bottom right-hand side). By increasing the bandwidth to 20 MHz, we obtain a predictive view of whether reliable communication is achievable within this frequency range and bandwidth. Please refer to Figure 6, the top right-hand side, for further details.

On the other hand, UWAC systems often utilise bandwidth that is relatively wide compared to the centre frequency of the signal, thus qualifying as wideband systems in a relative sense [24]. Therefore, we also investigate the behavior of the channel when employing a broader bandwidth with a high centre frequency for communication (see Figure 6, top left-hand side). However, it is widely recognised that the path loss of acoustic signals is frequency-dependent, and higher centre frequencies experience increased path loss, as depicted in Figure 6, bottom left-hand side, and Figure 6, top left-hand side.

In Figure 6, top left-hand side, only one or two dominant paths are visible and the delayed versions of channel responses are highly attenuated. In Figure 6, bottom left-hand side, we can observe multiple dominant paths along with several low-strength reflected micro-paths. These arise due to the narrowband characteristics of the channel, which leads to the presence of multiple unresolved micro-paths. Unresolved micro-paths result in frequency-selective destructive and constructive interference, although micro-paths themselves do not suffer from frequency-dependent attenuation.

Now, let us look again into the low-frequency responses in Figure 6 (bottom right-hand side) and Figure 6 (top right-hand side). Although the scenario presented in the bottom right-hand side has been experimentally validated as suitable for long-distance transmission underwater, this scenario suffers from a delayed version of reflected paths. Reflected paths can be strong in some instances, resulting in destructive interference; this necessitates the use of robust filtering on the receive side. However, based on the observation in the top right-hand side, a wider bandwidth for low-frequency range communications is recommended. In terms of the predictive behavior obtained from our model, a wider bandwidth can result in a lower number of sparse reflected paths and a strong dominant signal path.

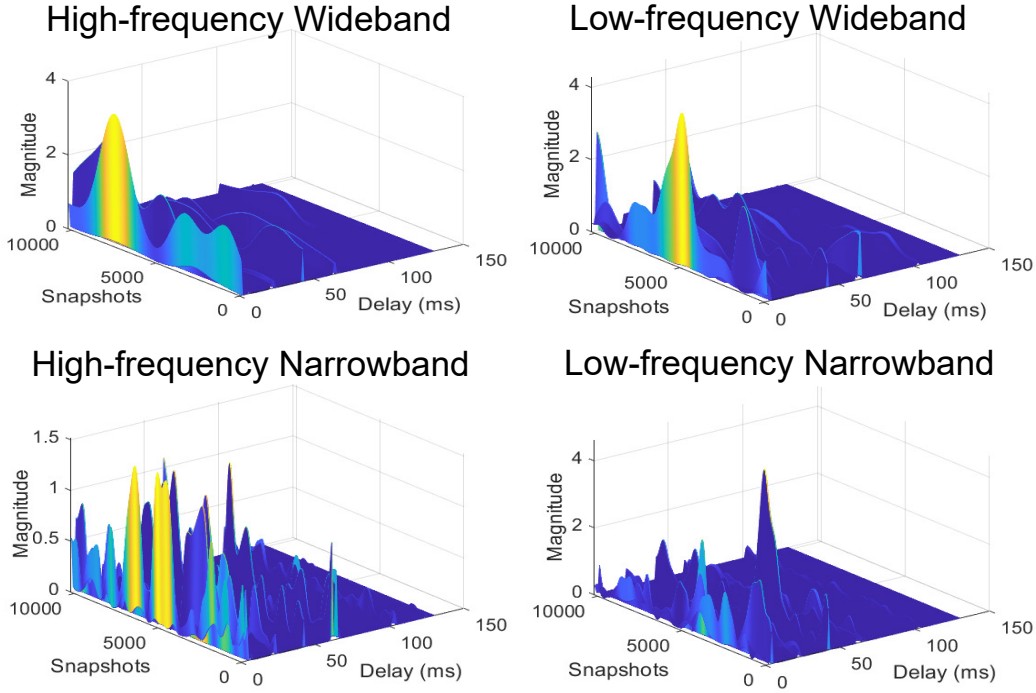

**Figure 6.** Generated channel samples using our formulated channel emulator platform with the possibility of wideband (20 MHz) and narrowband (10 KHz) acoustic signals travelling through water with high (100 MHz) and low (1 MHz) centre frequencies.

In the absence of actual measurement data, in order to establish the validity of our simulation platform, we compare our results with those generated by the well-established BELLHOP platform. We generated channel impulse responses (CIRs) using the BELLHOP simulator for four different transmitter depths transmitting hyperbolic frequency modulated pulses at centre frequencies of 4 kHz, bandwidths of 1 kHz, and pulse durations of 0.68 s (refer to Figure 7). A similar set of parameters is used to generate acoustic signal paths using our proposed simulator. Observing the positions of the direct path ($P_d$), the first surface reflection path ($P_s$), and the first bottom bounce path ($P_b$) on the delay axis, in the results generated, we can say that our proposed emulator platform reliably represents the UWAC scenarios.

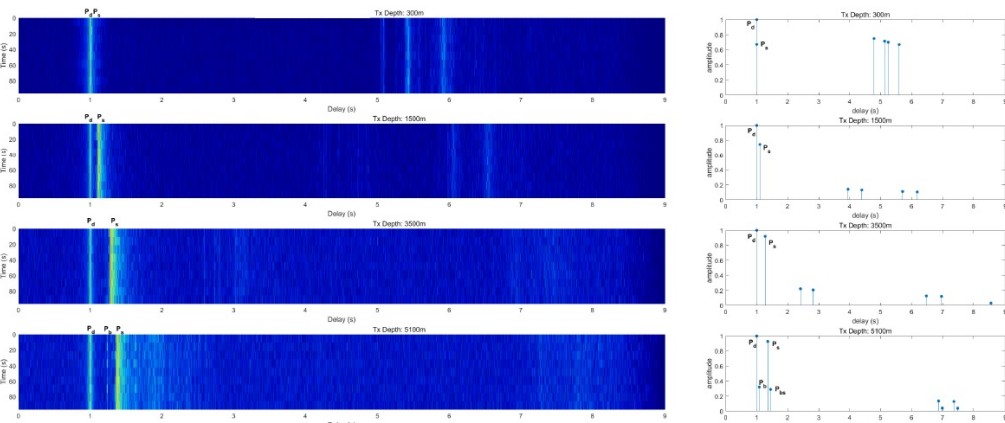

**Figure 7.** Generated channel samples using our formulated channel emulator platform (left-hand side) compared with the CIRs generated using the BELLHOP platform (right-hand side of the figure).

### 4.2. Possible Example Use-Cases

Here, we outline several ways in which the output of the channel emulator can be utilised to design reliable and efficient UWAC systems.

#### 4.2.1. Design of Probe Signal for Predicting System Performance

Let us represent the multi-tone signal used for channel sounding as

$$\tilde{x}(t) = \sum_{\hat{f}=\hat{f}_1}^{\hat{f}_W} \cos(2\pi \hat{f} \tilde{\delta} \hat{t} + \psi_{\hat{f}}) = \sum_{\hat{f}=\hat{f}_1}^{\hat{f}_W} \mathcal{R}\left\{ \tilde{X}_{\hat{f}} e^{j(2\pi \hat{f} \tilde{\delta} \hat{t})} \right\} \tag{30}$$

where $\tilde{\delta}$ is the frequency separation between tones, $\hat{T} = 1/\tilde{\delta}$ is the fundamental period of $\tilde{x}(t)$, $\hat{f}_1$ and $\hat{f}_W$ are the limits of the channel bandwidth and $\tilde{X}_{\hat{f}} = e^{j\psi_{\hat{f}}}$, where $\psi_{\hat{f}}$ is the relative phase of each tone; $\tilde{x}(t)$ corresponds to a Fourier series with complex coefficients $\tilde{X}_{\hat{f}}$. $\tilde{X}_{\hat{f}}$ can be designed using the Zadoff–Chu sequence as

$$\tilde{X}_{\hat{f}} = \exp\left( -j\frac{\pi \hat{u} \hat{f}^2}{W_{ZC}} \right) \tag{31}$$

where $W_{ZC}$ is the length of the Zadoff–Chu sequence and $\hat{u} \in \mathbb{W}$, such that $gcd(u, W_{ZC}) = 1$ where $gcd(\cdot)$ stands for the greatest common divisor function.

Next, the multi-tone signal $\tilde{x}(t)$ is communicated over our Simulink-based channel model. The received signal, in that case, is given by

$$\tilde{y}(t) = \int_0^\infty \tilde{x}(t - \tau_q) h(t, \tau_q) d\tau_q = \sum_{\hat{f}=\hat{f}_1}^{\hat{f}_W} \mathcal{R}\left\{ e^{j(2\pi \hat{f} \tilde{\delta} t + \psi_{\hat{f}})} \int_0^\infty h(t, \tau_q) e^{j(2\pi \hat{f} \tilde{\delta} \tau_q)} d\tau_q \right\}$$

$$= \sum_{\hat{f}=\hat{f}_1}^{\hat{f}_W} \mathcal{R}\left\{ H(t, \hat{f} \tilde{\delta}) e^{j(2\pi \hat{f} \tilde{\delta} t + \psi_{\hat{f}})} \right\} \tag{32}$$

where $H(t, \rho)$ is the time-varying channel frequency response.

Using the method outlined in (31) and (32), a multi-tone signal is first generated for probing the channel. The sampling frequency of both the transmitter and receiver is set to 30 kHz. The frequency spacing parameter $\tilde{\delta}$ determines the range of channel delays that can be estimated. In this case, $\tilde{\delta}$ is set to 100 Hz since the max delay spread of the simulated channel is 10 ms. The total bandwidth of the transmitted signal is $B = 4$ kHz, with the lower frequency bound $\hat{f}_L = 1$ kHz, and the upper frequency bound $\hat{f}_H = 5$ kHz. Accordingly, the number of tones is set to 40, and the phase parameter of each tone is generated by the

Zadoff–Chu sequence. Figure 8 shows the waveform and the frequency spectrum of the multi-tone signal. Next, the probe signal is sent through a channel generated using our Simulink model. A total of 100 groups of probe signals were sent (matching the number of snapshots of the simulated channel). Each probe signal contains a multi-tone signal with 3 periods, and its duration is 30 ms. A sample of the received signal is shown in Figure 9.

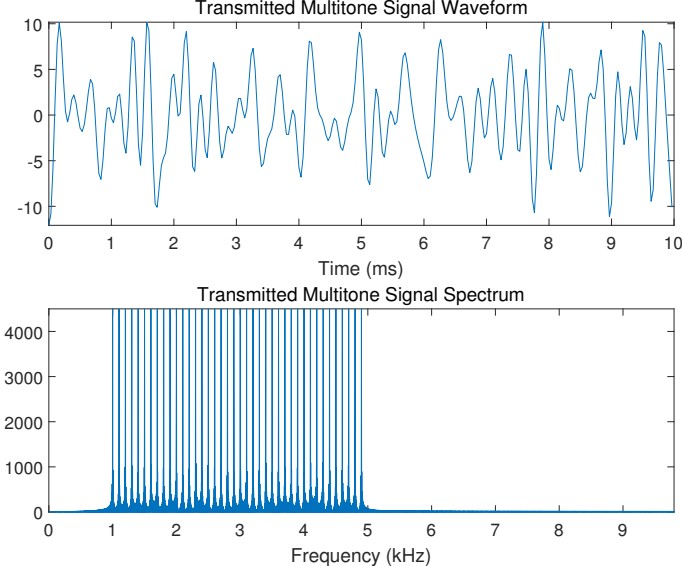

**Figure 8.** Transmitted signal waveform and spectrum.

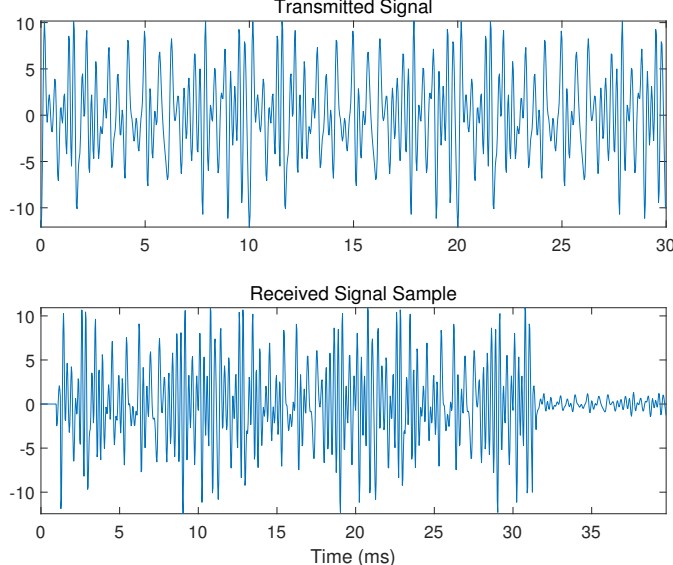

**Figure 9.** Received signal waveform.

### 4.2.2. Simulating System Performance over a UWAC Link

We begin by generating a sequence of pseudo-random binary numbers (PNs) through the use of a linear-feedback shift register (LFSR). The LFSR generates numbers with a value of either e +1 or −1, each with an equal probability. The PN sequence is then subjected to 3GPP Turbo coding [25] using a rate-1/2 Turbo code. Following this, the coded sequence is mapped onto a differential quadrature phase shift keying (DQPSK) constellation. The next steps involve computing the inverse fast Fourier transform (IFFT), appending a cyclic prefix (CP), and modulating the baseband sequence with the real part of a complex exponential carrier signal. The sequence is upsampled and multiplexed using orthogonal frequency

division multiplexing (OFDM). The resulting signal is transmitted through our Simulink-based channel model. On the receive side, the passband sequence is demodulated with a complex exponential carrier. The OFDM demodulator removes the CP, computes the fast Fourier transform (FFT), and demaps the received DQPSK sequence. It employs a soft decision-based demodulation approach using approximate log-likelihood ratios (LLRs). Approximate LLRs provide reliable results while avoiding not-a-number (NaN) errors that tend to occur at high signal-to-noise ratios (SNRs) when using exact LLRs. This is because exact LLRs involve computing exponentials of very large or very small numbers using finite-precision arithmetic, which ultimately results in high bit-error rates (BERs) [26] due to numerical instabilities.

Next, we simulate the BER performance of the above formulated (a) DQPSK-OFDM 1/2-rate Turbo coding for the single-transmit–single-receive system; (b) DMPSK-OFDM 1/2-rate Turbo coding for $4 \times 4$ multiple-transmit–multiple-receive system for four different use-case scenarios (i) SNR = 12 dB, message length = 1024, CP length = 64, roll-off factor of the transmit and receive filters = 0.2, sampling frequency = 300 Hz, upsampling factor = 24, $\chi = 0$, $K = 0$, $Q = 6$, $L = 500$, (ii) SNR = 12 dB, message length = 1024, CP length = 64, roll-off factor of the transmit and receive filters = 0.2, sampling frequency = 300 Hz, upsampling factor = 24, $\chi = 0.05$, $K = 0$, $Q = 6$, $L = 500$, (iii) SNR = 12 dB, message length = 1024, CP length = 64, roll-off factor of the transmit and receive filters = 0.2, sampling frequency = 300 Hz, upsampling factor = 24, $\chi = 0.05$, $K = 3$, $Q = 6$, $L = 500$, and (iv) SNR = 12 dB, message length = 1024, CP length = 64, roll-off factor of the transmit and receive filters = 0.2, sampling frequency = 300 Hz, upsampling factor = 24, $\chi = 1.2$, $K = 5$, $Q = 6$, $L = 500$. A comparative plot for BER curves is presented in Figure 10.

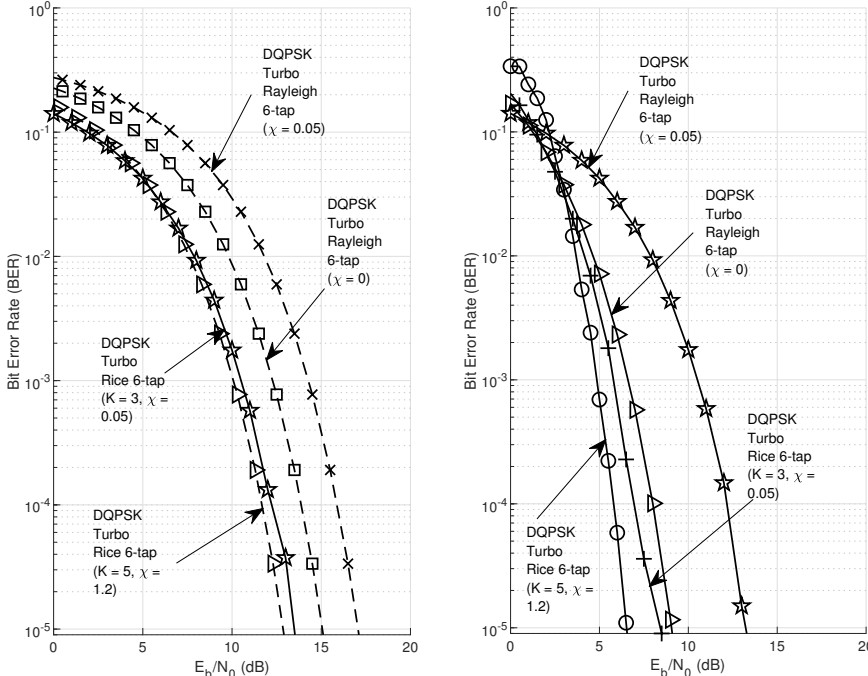

**Figure 10.** Comparative BER performances for the 6-tap channel model, where the curves are generated for DQPSK-OFDM 1/2-rate Turbo coding for the single-transmit–single-receive system (**left**-hand side); DMPSK-OFDM 1/2-rate Turbo coding for $4 \times 4$ multiple-transmit–multiple-receive system (**right**-hand side).

## 5. Conclusions

In this paper, a novel channel model is proposed for UWAC systems to describe the characteristics of the underwater acoustic channel. The model incorporates concepts such as damped harmonic oscillators, parametric oscillators, Doppler shifts and spreads, and SOS models. Based on the proposed model, a channel emulator was developed using the

Simulink platform and block sets, which could be used to generate practical UWAC links. The proposed emulator accounts for both time and delay domain variations in the channel and, consequently, will be able to offer realistic predictions of the performances of different channel estimations, precoding, and detection algorithms. The Simulink model can be easily extended to other platforms, coding techniques, and co-simulations for tractable hardware implementation. If needed, details on other physical processes, such as path loss, shadowing, surface motion, absorption, etc., can also be included in the proposed emulator by adding Simulink block sets.

**Author Contributions:** In this research article, I.D. and N.M. have contributed equally to the conceptualization and development of this work. All authors have read and agreed to the published version of the manuscript.

**Funding:** This material is based on work supported by Science Foundation Ireland (SFI) and is co-funded by the European Regional Development Fund under grant numbers 13/RC/2077 and 13/RC/2077-P2.

**Institutional Review Board Statement:** This study did not require any ethical approval.

**Informed Consent Statement:** This study does not involve humans.

**Data Availability Statement:** This is a theoretical study and no data is involved in generating the results.

**Conflicts of Interest:** The authors declare no conflict of interest. The funders had no role in the design of the study; in the collection, analyses, or interpretation of data; in the writing of the manuscript; or in the decision to publish the results.

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
