# Peer review of "Channel Emulator Framework for Underwater Acoustic Communications"

_applsci, doi:10.3390/app13095818_

Round 1

Reviewer 1 Report

Major comments

  1. The manuscript needs to include how the measurement campaign details helped to relate the mathematical analysis and the measured data. 
  2. The measurement systems configurations and block diagram can help readers understand the measurement scenario.
  3. How do the authors include different water-body and geographical dependencies in the Simulink model?
  4. In the contribution section, the authors have mentioned that they developed a channel emulator platform, demonstrating (a) snapshots of channel samples for different scenarios and parameters and (b) the performance of two example communication systems. However, the details of the example system need to be included in the later part of the manuscript.
  5. Figure 6 deserves more description to understand by its readers. 
  6. How does the last sentence of the abstract support the presented results? The authors must clarify and support every claim by the presented results.

Minor comments

Please allow space after the number to type units and follow the standard rules to write the unit abbreviations.

Author Response

We have provided point-by-point response to the concerns of the Reviewer in the attached file.

Reviewer 2 Report

Attached Separately.

Author Response

(The authors gave the same response as above.)

Reviewer 3 Report

This paper proposed a mathematical model and emulator framework involving the damped harmonic oscillators and Milne’s oscillator technique for underwater acoustic communications. The proposed model and framework is mathematically analyzed and some channel samples are generated via the proposed framework. However, some concerns should be addressed before its publication.

1. In the abstract and introduction parts, the authors should address the characteristics of the proposed model and the emulator framework, especially compared to the existing schemes.

2. The variables in equations should be carefully explained, such as those in equation (2).

 3. In the equation analysis, how to insert (5) in (2), as they have different variables x and y.

4. In the numerical results section, the authors should explain why these three different scenarios are selected.

 5. In the results and discussion section, the author should more specific description for Block diagram of the Simulink model for the channel emulator as in Fig. 5.

 6. In the results and discussion section, the author should more discussion for the generated channel samples as in Fig. 6. 

Author Response

(The authors gave the same response as above.)

Round 2

Reviewer 1 Report

The authors have improved the manuscript compared to the previous version. I recommend accepting the manuscript.

Reviewer 3 Report

All my concerns have been addressed.